# An improved biometric stress monitoring solution for working employees using heart rate variability data and Capsule Network model

**Mashael M. Khayyat**[1], **Raafat M. Munshi**[2], **Bayan Alabduallah**[3], **Tarik Lamoudan**[4], **Ehab Ghith**[5], **Tai-hoon Kim**[6]*, **Abdelaziz A. Abdelhamid**[7,8]*

1 Department of Information Systems and Technology, College of Computer Science and Engineering, University of Jeddah, Jeddah, Saudi Arabia, 2 Department of Medical Laboratory Technology (MLT), Faculty of Applied Medical Sciences, King Abdulaziz University, Rabigh, Saudi Arabia, 3 Department of Information Systems, College of Computer and Information Sciences, Princess Nourah bint Abdulrahman University, Riyadh, Saudi Arabia, 4 Department of Mathematics, College of Science and Arts, Muhayil, King Khalid University, Abha, Saudi Arabia, 5 Department of Mechatronics, Faculty of Engineering, Ain shams University, Cairo, Egypt, 6 School of Electrical and Computer Engineering, Yeosu Campus, Chonnam National University, Yeosu-si, Jeollanam-do, Republic of Korea, 7 Department of Computer Science, College of Computing and Information Technology, Shaqra University, Shaqra, Saudi Arabia, 8 Department of Computer Science, Faculty of Computer and Information Sciences, Ain Shams University, Cairo, Egypt

* taihoonn@chonnam.ac.kr (TK); abdelaziz@su.edu.sa (AAA)

**Data Availability Statement:** The complete dataset citation reference is added in the dataset section of the paper. Furthermore, the direct link of publicly

## Abstract

Biometric stress monitoring has become a critical area of research in understanding and managing health problems resulting from stress. One of the fields that emerged in this area is biometric stress monitoring, which provides continuous or real-time information about different anxiety levels among people by analyzing physiological signals and behavioral data. In this paper, we propose a new approach based on the CapsNets model for continuously monitoring psychophysiological stress. In the new model, streams of biometric data, including physiological signals and behavioral patterns, are taken up for analysis. In testing using the Swell multiclass dataset, it performed with an accuracy of 92.76%. Further testing of the WESAD dataset reveals an even better accuracy at 96.76%. The accuracy obtained for binary classification of stress and no stress class is applied to the Swell dataset, where this model obtained an outstanding accuracy of 98.52% in this study and on WESAD, 99.82%. Comparative analysis with other state-of-the-art models underlines the superior performance; it achieves better results than all of its competitors. The developed model is then rigorously subjected to 5-fold cross-validation, which proved very significant and proved that the proposed model could be effective and efficient in biometric stress monitoring.

## 1 Introduction

Stress is one of the most widespread factors contributing to both physical and mental health issues [1]. However, in contemporary circumstances, with high pressure at work, the

available dataset is: https://www.kaggle.com/datasets/qiriro/stress.

**Funding:** This research work was funded by Institutional Fund Projects under grant no. (IFPIP: 1446-415-1443). The authors gratefully acknowledge technical and financial support provided by the Ministry of Education and King Abdulaziz University, DSR, Jeddah, Saudi Arabia. This study was also supported by Princess Nourah Bint Abdulrahman University Researchers (Supporting Project number PNURSP2024R440), Princess Nourah Bint Abdulrahman University, Riyadh, Saudi Arabia. The authors extend their appreciation to the Deanship of Scientific Research at King Khalid University for funding this work through large group Research Project under grant number RGP 2/218/45. The authors would be happy to thank the Deanships of Scientific Research at Shaqra University for supporting this work.

**Competing interests:** The authors have declared that no competing interests exist.

establishment of effective stress management techniques in the workplace appears to be highly relevant both for employee health and productivity. The classic approaches to assessing stress are mainly subjective and based on self-reports. Heart Rate Variability (HRV) provides a robust, objective measure of stress by reflecting autonomic nervous system activity. This paper provides an advanced solution for biometric stress monitoring, using HRV data and a Capsule Network model that outperforms traditional Convolutional Neural Networks (CNNs) in complex pattern recognition. Our approach is to real-time and accurate stress detection, which will facilitate timely interventions to foster a healthier workplace. Defined as the organism's response to internal or external stimuli, stress incorporates a spectrum of experiences, from functional adaptive responses to difficult circumstances to hazardous overload [2]. An inherent mechanism acts upon the body as it always strives to regain its balance through adverse conditions. Stress-related conditions are identified as one of the most common health problems [3], and are estimated to account for a large percentage of all medical visits in both Europe and the United States. The problem consequently involves a significant burden on healthcare systems [4].

The initial phase of stress occurs when an organism encounters stimuli or circumstance referred to as stressors [3]. These stressors manifest in various forms, broadly classified as psychological and physiological. Psychological stressors include situations such as financial debt, bereavement, unemployment, academic pressures, and similar challenges. On the other hand, physiological stressors comprise factors such as infections, extreme temperatures, and inadequate relaxation. Upon perceiving a stress-inducing situation, the body initiates short-term or long-term responses. Central to this process is the hypothalamus, a vital brain region that orchestrates the stress response. It activates the pituitary gland, prompting the release of cortisol from the adrenal gland. Cortisol aids in stabilizing blood sugar levels and restoring normal bodily functions. Currently, the adrenal medulla, part of the autonomic nervous system, is activated by the hypothalamus to produce rapid stress reactions. This leads to the secretion of adrenaline, triggering the fight-or-flight response and activating the sympathetic nervous system. Once the stressor diminishes and the parasympathetic nervous system assumes control, the body returns to its baseline state [5].

Stress can be categorized into three distinct types, each exhibiting unique symptoms, characteristics, durations, and treatment approaches. The first type is acute stress, which is the most prevalent and is characterized by a short duration often associated with negative thoughts. Episodic stress refers to continuous, intense stress experienced over a prolonged period, often becoming habitual. The last among the three kinds of stress is chronic stress, which may have its roots in early childhood or some other traumatic experiences that have left an indelible mark on a person's life [6].

Stress is a complex condition that affects both old and young alike. The pressure on the employees is increasing every day, making the workplace one of the significant stressors in recent times [7]. The resources necessary for employees to perform their jobs may be unavailable or inadequate, and their needs may not be properly addressed. Work-related stress has been associated with high absenteeism, errors, and low productivity [8]. In fact, social welfare, health care, and the alleviation of programs for stressed, burned-out, or depressed workers are estimated to cost the European Union 617 billion euros every year, a powerful indication that workplace stress [9], is not only an issue between a worker and his boss, rather, it affects the larger community. The most common form of pressure experienced by many adolescents is academic stress, which is mental pressure obtained from various expectations. Managing stress can be challenging, as students face various demands, including homework, exams, coursework, social interactions, and familial responsibilities, all of which directly influence their

academic performance. High levels of stress among students often coincide with symptoms of depression and anxiety [10].

Extensive research has established a clear link between elevated stress levels, decreased well-being, and a lower quality of life. Prolonged exposure to stress can precipitate serious mental health conditions such as anxiety and depression [11]. A comprehensive survey involving 5,551 students [12] revealed a negative correlation between anxiety levels and academic achievement, indicating that students experiencing lower anxiety levels tend to achieve higher GPAs compared to those with moderate or high anxiety levels. In addition, the impact of depression and anxiety can escalate to the point of suicide, which ranks as the second leading cause of death among college and university students. Reports indicate that approximately 1,100 students per 100,000 commit suicide annually [13]. Monitoring stress levels can prove invaluable for universities and families, facilitating better support for student's academic success and overall well-being.

The utilization of recent technological tools and methodologies within the emerging domain of affective computing has demonstrated significant potential in automatically monitoring and detecting occupational stress. Key physiological signals and measurements, including electrocardiogram characteristics, electrodermal activity, skin temperature, and electromyographic activity, have been extensively researched and validated as reliable indicators of stress. This study leverages these signals with a transformer model to detect stress levels among individuals, showcasing its efficacy for effective stress management and coping strategies. The main contributions of this study are

- To enhance the predictive accuracy of biometric stress monitoring, a novel CapsNets model is introduced. The proposed model is tested on two independent benchmark datasets SWELL and WESAD for performance investigation.

- The study involves an assessment of the performance of established deep transfer learning algorithms applied to biometric stress monitoring data. These algorithms encompass Xception, EfficientNetB4, CNN, VGG19, ResNET, MobileNet, and InceptionV3.

- The effectiveness of the proposed approach is thoroughly examined through extensive experiments, and a comparative analysis with various state-of-the-art methods is conducted. To validate the robustness of the proposed approach, the results are further substantiated using k-fold cross-validation.

The paper is structured to provide a comprehensive exploration of stress monitoring using machine learning methodologies and biometric signals. Section 2 delves into a detailed literature review, analyzing existing approaches that utilize various biometric signals for stress monitoring within the context of machine learning. Moving forward, Section 3 outlines the experimental protocol, elucidating the machine learning approach adopted and the systematic procedure employed for network development. Subsequently, in Section 4, the paper presents statistical findings derived from the experimentation process, critically evaluates the effectiveness of the proposed network, and conducts a comparative analysis with established benchmark machine learning models. Finally, Section 5 offers conclusive remarks, discussing potential limitations of the study and proposing avenues for future research in this domain.

## 2 Related work

While the fundamental understanding of stress as a psychological phenomenon is well established, its practical application remains challenging due to its highly individualized nature. Modern technologies for stress detection have advanced to address multiple factors and their

interconnected causal relationships that contribute to stress. This section introduces various existing methods for identifying and analyzing stress states, all grounded in the analysis of biometric data.

Kizito Nkurikiyeyezu et al. [14] propose a person-specific biometrics generic stress system with a simple yet very effective calibration methodology that extracts accurate and personalized models for stress prediction from physiological samples collected from a large population. The authors have validated this approach on two stress datasets and demonstrated its superiority over a generic model. For example, while the generic model realizes an accuracy of only 42.5% ± 19.9%, their system achieved an accuracy as high as 95.2% ± 0.5% with calibration with just 100 samples. Another contribution by Kim et al. [15] focuses on the identification of child stress states based on biometric information in mobile environments. They classify a child's state of stress into four classes using normalized voice and heart rate data to perform classification and then evaluate the system's reliability. This study applies standard classification models, such as NB, DT, and SVM, to machine-learning-based biosignal analysis. In the experiment with voice and heart rate data, this resulted in an accuracy of 65.88%, 87.32%, and 88.53% for NB, DT, and SVM models, respectively; thus, the best-performing model was SVM.

Kenneth Lai et al. [16] present a prototype of an Intelligent Stress Monitoring Assistant (SMA), designed as the next generation of stress detectors tailored for first responders. The SMA integrates, a residual-temporal convolution network for data learning from sensors and stress feature detection, and (b) a reasoning mechanism based on a causal network for fusion at multiple levels. They evaluated SMA using the WESAD multifactor physiological dataset and achieved an accuracy of 86% for stress recognition and an impressive 98% for stress detection. In a related study, Bosun Hwang et al. [17] propose an optimal architecture employing recurrent and convolutional neural networks for stress monitoring using ultra-short-term ECG signals. Their system, Deep ECGNet, determines optimal convolution filter length and pooling length through optimization experiments and waveform characteristic analysis of ECG signals. The deep ECGNet framework surpasses conventional methods, achieving the highest accuracy of 87.39% in recognizing stress conditions.

Seo et al. [18] proposed a deep end-to-end stress detection algorithm using various physiological signals, including electrocardiogram and respiration signals. The experiment was conducted using tasks that included the stress-inducing Stroop and math tasks at the workplace scenario, followed by a relaxation task. The results of this study prove that the proposed network has massive efficiency, with the achieved average accuracy reaching 83.9% and its average F1 score being 81%. In a related study, Mariano et al. [19] propose an AI-based stress detection system using heart rate data and conduct their analysis along two datasets: WESAD and SWELL-KW. The authors used LOF and MLP to detect stress using heart rate information. The results underscore the supremacy of the MLP model in achieving the highest accuracy scores of 99.04% on WESAD and 88.64% on the SWELL dataset.

Koldjik et al. In [20] have conducted studies developing automated classifiers to find out the working conditions and stress-related mental state from a multimodal non-invasive sensor dataset recently. This dataset included physiological measures, such as heart rate and skin conductance, as well as behavioral measures, like facial expressions, body posture, and computer interaction. It turned out that, unlike other machine learning classification methods, SVM could recognize neutral and stressful work conditions with 90% accuracy, even without including computer interaction features among the most informative indicators. In contrast, researchers in [21] created individual models for stress classification based on ANNs and stated that the most reliable indicator of stress is body posture. They applied early and late fusion techniques on multimodal data fusion and recorded an accuracy as high as 96%.

**Table 1. Literature review analysis table.**

| Ref | Classifier | Dataset | Performance |
|---|---|---|---|
| [14] | Hybrid Calibration method, Generic models, Person-specific models | SWELL, WESAD | 95.02% Hybrid Calibration Method |
| [15] | NB, DT, SVM | Self-collected (voice and heart rate data) | 88.53% SVM |
| [16] | SMA | WESAD | 86% stress recognition, 98% stress detection |
| [17] | kNN, LSVM, RBFSVM, DT, RF, MLP, ADA, NB, QDA, Gaussian Process, Deep ECGNet | Kwangwoon university | 87.39% Deep ECGNet |
| [18] | RF, DT, SVM,KNN, DeepER Net (CNN and LSTM) | 16 different datasets | 83.9% DeepER Net |
| [19] | LOF, MLP | SWELL, WESAD | 99.04% MLP on WASAD, 88.64% MLP on SWELL |
| [20] | KNN, SVM, Bayesian Approaches, ANN, Classification tress | SWELL | 90% SVM |
| [21] | ANN | SWELL | 96.09% |
| [22] | SFSS, RSFS, SFFS+GDA, SBS, MERNN | WESAD | 92.43% MERNN |
| [23] | RF, SVM, LR | Empatica E4 device self-collected dataset | 76.5% RF |

Vellaisamy & Freitas [22] proposed an optimized feature selection model and a classifier to improve stress detection in ECG signals. They have proposed the African vulture optimization metaheuristics model for FS that will select only relevant features, hence reducing data for classification. An AVO-based modified Elman recurrent neural network technique is proposed for effective classification where AVO is used to optimize the weights of MERNN. The experimental results indicate that the MERNN could be as accurate as 92.43%. Another work by Campanella et al. [23] proposes a system to analyze the signals coming from the Empatica E4 bracelet, enhanced by three different ML models: random forest, SVM, and logistic regression. They extract 27 characteristics from photoplethysmographic and electrodermal activity signals from 29 subjects for binary classification. The results show that the RF model yields an accuracy of 76.5% when using all features. This large volume of studies has ranged from selecting and classifying with optimized features in ECG signals to detecting their stressors using machine learning models with wearable device signals. The comprehensive analysis of related work is shown in Table 1.

## 3 Material and methods

This section details the proposed methodology for stress classification, along with an overview of the stress monitoring dataset. We describe the architecture of the CapsNets model and its components that contribute to efficient classification. The complete workflow methodology of the proposed model is shown in Fig 1. The details of hyperparameters for all the learning models used in this work are shown in Table 2.

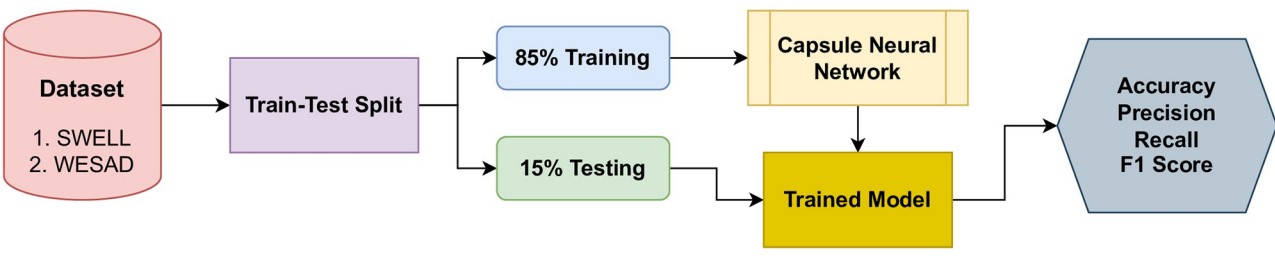

**Fig 1. Proposed methodology diagram.**

**Table 2. Hyperparameters for various transfer learning models.**

| Model | Learning Rate | Batch Size | Epochs | Optimizer |
|---|---|---|---|---|
| Xception | 0.001 | 32 | 50 | Adam |
| EfficientNetB4 | 0.001 | 32 | 50 | Adam |
| CNN | 0.01 | 64 | 100 | SGD |
| VGG19 | 0.0001 | 32 | 50 | Adam |
| ResNet | 0.001 | 32 | 50 | Adam |
| MobileNet | 0.001 | 32 | 50 | Adam |
| InceptionV3 | 0.001 | 32 | 50 | Adam |
| Capsule Network | 0.0005 | 32 | 50 | Adam |

## 3.1 Transfer learning models for stress monitoring

**3.1.1 Xception.** In the dynamic scene of stress monitoring by using biometric methods, adopting advanced machine learning models like Xception has proven to be a game changer [24]. Xception, with its profound learning ability and hierarchical feature extraction, comes up with a paradigm shift in understanding and analyzing biometric data related to stress responses. Recent research has underlined the potential of Xception for the processing of a wide variety of biometric signals, from ECG reading to galvanic skin response data.

Researchers have also uncovered subtle physiological markers indicative of stress using the complex layers in the Xception architecture. This forms one of the critical strengths of Xception in capturing spatial and temporal dependencies within biometric data that detect subtle variations associated with stress. This granularity identifies stress and allows differentiation across different intensities in stress, instilling a deeper understanding of individual stress dynamics. The versatility of Xception extends to multimodal biometric fusion, wherein facial expressions, voice patterns, and HRV can be combined and evaluated integrally. In this way, a more global picture of an individual's stress profile is obtained, helpfully for both personalized interventions and targeted stress management strategies. With Xception, this generalization extends to real-life applications, ranging from wearable devices for continuous monitoring to clinical setups for retrospective analysis. It can be further scaled and adapted for stress monitoring frameworks so that researchers and practitioners may use Xception at the leading edge of tapping into state-of-the-art technology for increasingly relevant stress management challenges. During these high-pressure times, the interplay between biometrics and deep learning through models like Xception empowers us to be at the forefront of the newest research in proactive stress monitoring and intervention. We continue to innovate and integrate state-of-the-art methodologies, stepping toward a future where stress management is not just reactive but preemptive—empowering all individuals to be healthy and resilient.

**3.1.2 EfficientNetB4.** In this research into measuring stress with biometric methods, we have applied cutting-edge EfficientNetB4 as a central component in our analysis [25]. That is to say, the EfficientNetB4 model was known to perform well in all image recognition tasks, and thus, using it became instrumental in detecting and interpreting biometric indicators of stress. Enriched with the power of the EfficientNetB4 model, it has been possible to extract many intricate features from biometric data, such as facial expressions, skin conductance responses, and heart rate variability. This helped us in delineating the minute-scale physiological and ocular cues related to stress; this model perceives valuable information about an individual's emotional well-being. Efficiency and scalability have provided real-time analysis through the EfficientNetB4 model, facilitating timely interventions and personalized feedback. This model has efficiently processed complex visual and physiological information to a large

extent, improving the accuracy and effectiveness of our stress detection system. It is owing to the advanced version of the EfficientNetB4 model that our research in the monitoring of stress through biometric methods could be taken several steps ahead. It has pioneered image recognition and feature extraction toward more accurate and insightful valuations of stress, leading to greater welfare and health.

**3.1.3 Convolutional neural network.** It infers knowledge from various articles and research findings on CNN models. CNNs are potential tools in this regard by proving their worth in the analysis of biometric data for stress pattern detection [26]. Much research is done on CNN architectures for stress monitoring with a primary focus on inputs like HRV, skin conductance, facial expressions, and voice features. Such applications of CNN models in stress monitoring exploit their deep learning capabilities for feature extraction and pattern recognition. In essence, it triggers the development of algorithms that can precisely decipher biometric signals that indicate responses to stress. It has been reported that such a model realized enormous successes in the accurate categorization and subsequent monitoring of stress, having achieved its training using labeled datasets that contain diversified data points on stress. One of the striking advantages associated with using a CNN model in stress monitoring is its capability to process complex structures of data and abstractions of relevant features automatically. This is quite important for the analysis of subtle changes in biometric signals that indicate varying levels of stress. Further, CNNs are connected with wearable biometric devices, facilitating data acquisition on a real-time basis to track changes in stress levels. However, addressing data variability, model interpretability, and generalization should be key areas of future research. This will further enhance the effectiveness and reliability of CNN-based stress monitoring systems even more. Nevertheless, the adaptation of CNN models represents an outstanding move toward the future in this field of study, considering the airway identified is toward robust and accurate biometric solutions for stress monitoring.

**3.1.4 Visual Geometry Group (VGG19).** Advanced machine learning models, for example, VGG19, have been one of the avenues of interest. VGG19 is a convolutional neural network architecture that has already been exploited as very effective in extracting complex features from biometric data and served to deepen understanding of the stress responses [27]. New research has explored the use of VGG19 for the analysis of physiological signals, including HRV, electrodermal activity, and facial expressions. These signals may be fed through the layers of VGG19 to reveal subtle patterns or nuances correlating with different stress levels. Now, one key strength for VGG19 would be its capacity for learning hierarchical representations that unravel minute changes within biometric data, which would be obscured otherwise by more traditional analysis techniques. This model thus enables the estimation of stress and temporal dynamics, opening up perspectives for individual interventions in stress management. The flexibility of VGG19 across various biometric modalities makes it versatile in the framework of stress monitoring. From real-time monitoring in wearables to retrospective analysis in clinical settings, there seems to be no bind to the ability of VGG19 to foster innovation for biometric-driven assessment of stress. As we journey through modern life's labyrinthine challenges, the fusion of biometrics with deep learning through models such as VGG19 is heralding a new era in monitoring stress. With that comes the power of data-driven insight bringing one closer to a place where proactive stress management will no longer be a possibility but a reality.

**3.1.5 Residual networks.** Residual networks, also known as ResNet, have become a viral residual network for the extraction of intricate features from biometric data and offer holistic insight into stress responses. Recent research has proved the effectiveness of ResNet in analyzing several biometric signals, such as ECG data, skin conductance, and facial expressions [28]. It is in this multi-layered arrangement that researchers, armed with ResNet, have been able to

discover the subtle patterns and correlations within this biometric data relating directly to levels of stress. One significant advantage of ResNet is mitigating a common issue during training the vanishing gradient problem in deep neural networks. This permitting resilience lets ResNet capture the temporary dynamics and subtle changes in the biometric signals to make a much more fine-grained assessment of the stress levels and patterns. Further, it provides flexibility regarding fusion with several biometric modalities, including physiological data associated with either audio or visual cues. This will provide a mosaic approach to not only yield higher accuracy in the detection of stress but will also help formulate personalized strategies for managing stress based on individual needs. In these respects, ResNet has practical stress monitoring applications in the real world, from wearable devices for continuous tracking to clinical environments for longitudinal studies. Adaptability and scalability are valuable virtues that ResNet brings to bear in deciphering the complexities of stress dynamics for effective interventions. This will be an essential step toward integrating ResNet into biometric-driven methodologies for stress monitoring as we navigate life's challenges. It is in this capacity that ResNet can be used to help researchers and practitioners understand the intricacies of stress responses, hence setting a clear path for proactive stress management and improved well-being.

**3.1.6 Capsule Networks (CapsNets).** Capsule Networks, commonly called CapsNets, is a novel neural network architecture propounded by Geoffrey Hinton and his colleagues [29] in 2017 as an alternate line of thinking to traditional convolutional neural networks. CapsNets are trying to capture some limitations with conventional CNNs in handling hierarchical relationships in data. A capsule in CapsNets is simply some general unit, like scalar output activations in CNNs, representing instantiation parameters of a related entity, for example, an object or part of it. The capsules are called "activations," as they output vectors that give the presence and properties of entities represented in the input data. CapsNets are designed to better extract the hierarchical structures in data than CNNs and hence apply to various tasks, from image classification to object detection and natural language processing using capsules.

A key feature of Capsule Networks is dynamic routing, where capsules communicate to reach a consensus on the presence of entities in the input data [30]. Dynamic routing updates the linkage coefficients between capsules iteratively based on an agreement of their prediction and input data so capsules can achieve consensus on the presence and features of entities. The active routing process in CapsNets is a suitable mechanism for handling changes in pose, viewpoint, and deformation of objects, which are hostile to traditional CNNs. In addition, due to their design capability in capturing hierarchical relationships in data, CapsNets have shown encouraging results for tasks such as image classification with small-sized datasets and robustness to various adversarial attacks.

Several research studies have been made on the applications and extensions of Capsule Networks to various other domains [31]. CapsNets have already been extended to object detection, segmentation, and pose estimation tasks, which prove quite effective in handling complex underlying data structures. Though relatively new, capsule networks remain an active area of research, with many efforts underway to enhance their efficiency, scalability, and applicability to various machine learning and artificial intelligence tasks.

**3.1.7 InceptionV3.** InceptionV3 is also a CNN architecture; it is an instance of an initially designed image recognition model found in new uses for healthcare, particularly in stress monitoring. Its ability to extract intricate features from complex data makes it suitable for analyzing physiological signals indicative of stress [32]. Biometric signals, like HRV, EDA, and facial expressions, are information-rich sources portraying an individual's stress response. These signals can be fed into the InceptionV3 model, enabling it to be trained on pattern recognition correlated with different stress levels. One of the most essential advantages of InceptionV3 is its pre-trained weights on large datasets like ImageNet [33]. This pre-training

enhances the model in generalization and extracting meaningful features from diverse bio-metrics data. Fine-tuning the model with domain-specific data further improves the model's accuracy in stress classification tasks. Preliminary research into using InceptionV3 for stress detection has shown promising results. Changes in HRV patterns, skin conductance levels, and facial microexpressions could infer a person's stress state with an excellent degree of accuracy when monitored continuously. This not only constitutes a non-invasive strategy with real-time feedback but opens ways to individualized interventions related to managing stressful conditions. Deep learning techniques, especially the InceptionV3 model, have high scalability and can thus be put inside wearable devices and even mobile applications. This will enable the continuous monitoring of stress levels in everyday settings and empower individuals with information to act for their well-being. Incorporating the InceptionV3 model within biometric stress monitoring systems enables new opportunities in personalized health. The robustness, adaptability, and accuracy of the model give promising prospects for furthering stress management strategies and promoting mental wellness at large.

### 3.2 Evaluation parameters

This work will resort to a variety of performance criteria to assess the goodness of fit for the proposed method while predicting biometric stress [34]. Amongst them, typical parameters like accuracy, F1 score, recall, and precision are usually used within machine learning to evaluate the performance achieved by any model. The latter works according to the following metrics:

- **True Positive (TP)**: A TP is where the ML model correctly predicts a positive instance as positive. This is how many of the positive outcomes the model identified correctly.

- **True Negative (TN)**: A TN represents when the model correctly predicts a negative instance as unfavorable. It shows the number of adverse outcomes that the model identified correctly.

- **False Positive (FP)**: An FP is when the ML model misclassifies a negative instance as positive. It might be interpreted as the number of errors the model made due to over-prediction of the positive outcome when it is actually harmful.

- **False Negative (FN)**: When the ML model mistakenly predicts a positive event as negative. It tells how many times the model was wrong by predicting the negative class when it was actually positive. These terms are used along with performance metrics such as the F1 score, recall, accuracy, and precision to measure the level of performance by the model.

One of the most important and commonly used parameters to evaluate the efficiency of models is accuracy. It refers to the percentage of correctly predicted occurrences to all expected instances. It can be calculated using the following formula:

$$Accuracy(A) = \frac{TP + TN}{TP + TN + FP + FN} \tag{1}$$

Precision is the ratio of the number of correctly predicted positive cases to the total positive cases. It can be computed with the following formula:

$$Precision = \frac{TP}{TP + FP} \tag{2}$$

Recall measures the completeness of the classifier. It shows the ratio of correctly detected actual positive cases. It is calculated as:

$$Recall = \frac{TP}{TP + FN} \tag{3}$$

The F1 score is considered a balanced and representative performance of the model. It includes both the accuracy and the recall of this model. Mathematically, the F1 score can be regarded as the harmonic average of recall and precision. This can be computed using.

$$F1 - Score = 2 \times \frac{Precision \times Recall}{Precision + Recall} \tag{4}$$

The Area Under the Curve (AUC) can be computed using the trapezoidal rule, which is a numerical method to estimate the integral of a function. To calculate the AUC using the trapezoidal rule, the area under the ROC curve can be approximated as follows:

$$AUC = \sum_{i=1}^{n-1} \frac{(x_{i+1} - x_i)(y_{i+1} + y_i)}{2} \tag{5}$$

$$= \sum_{i=1}^{n-1} \frac{\Delta x_i (y_{i+1} + y_i)}{2} \tag{6}$$

$$= \sum_{i=1}^{n-1} \left(\frac{\Delta x_i}{2}\right)(y_{i+1} + y_i) \tag{7}$$

where:

- $x_i$ and $x_{i+1}$ are consecutive points on the x-axis (False Positive Rate).

- $y_i$ and $y_{i+1}$ are the corresponding points on the y-axis (True Positive Rate).

- $\Delta x_i = x_{i+1} - x_i$ is the width of each trapezoid.

## 4 Experiments and analysis

### 4.1 Experimental setup

The "Biometrics for stress monitoring" dataset is downloaded from Kaggle, a website hosting benchmark datasets. This dataset integrates electrodermal activity (EDA) and heart rate variability (HRV) information taken from the SWELL and WESAD datasets [20, 35]. It is structured into three main directories and sub-directories for efficient organization and analysis. The "interim" directory contains transformed intermediate data, including ground truth labels (Labels), raw EDA signals (eda), and inter-beat intervals (ibi) derived from ECG signals. The "processed" directory houses files computed from the intermediate data, aiding in analysis procedures. Within the "final" directory, there are two subdirectories: "results," which contains specific published outcomes from the aforementioned paper, and "datasets," which contain combined train, test, and validation data used for model creation. These subdirectories are further detailed in the paper [14]. This well-structured setup makes it easy to share and use the dataset for research purposes in stress prediction models.

The Python 3.9 programming environment is used to conduct the research. The study's experimental setting includes the computer language Python 3.8, (Scikit learn version Version

1.5.0 and TensorFlow version r2.15), RAM capacity available (8GB DDR4), operating system type (64-bit Windows 11), CPU specifications are Intel Core i7 with a processor frequency at about 2.8 GHz which belongs to the 7th generation and an Nvidia GTX1060 GPU. This information is relevant for comprehending the technical characteristics of the research setting and the computational resources employed in this study.

## 4.2 Result of models on Swell dataset

In the first phase of the experiment transfer learning models and the proposed CapsNets model is applied to the Swell dataset, which has 3 classes, 'no stress', 'time pressure', and 'interruption'. Results of the learning models on the Swell dataset are given in Table 3.

Several ML models are compared in the table according to performance criteria including accuracy, precision, recall, and F1 score. Among the models evaluated, the CapsNets exhibited the highest accuracy of 92.76%, accompanied by impressive precision, recall, and F1 score values of approximately 91-92%. This model stands out for its robust predictive capabilities across various classes. The MobileNet model secured the second place with an accuracy of 91.62%, recall, precision, and F1 score of about 89%, showcasing its effectiveness in classification tasks. InceptionV3 and VGG19 also demonstrated good performance, with accuracy scores of 90.70% and 90.09%, respectively. However, their precision, recall, and F1 score metrics were slightly lower, ranging from 83% to 89%. On the other hand, models like Xception, EfficientNetB4, CNN, and ResNET showcased decent accuracy scores in the range of 86% to 83%, with corresponding precision, recall, and F1 scores around 80% to 85%. These results offer insightful information on the comparable performance of these ML models, highlighting the strengths and areas for improvement in their predictive capabilities.

## 4.3 Result of models on WESAD dataset

Another dataset used for experiments is the WESAD dataset which includes psychological and acceleration signals. This dataset also has 3 classes which are 'baseline condition', 'amusement condition', and 'stress condition'. Results of the proposed approach and other models on the WESAD dataset are given in Table 4.

Table 4 provides an extensive analysis and comparison of several ML models according to performance parameters including F1 score, accuracy, precision, and recall. Among these models, the CapsNets continues to exhibit exceptional performance, achieving a remarkable accuracy of 96.76% along with high precision, recall, and F1 score values around 97% to 98%. This proves its high predictive powers across different data sets. In the immediate following, ResNET and MobileNet are chalking significant improvements of 94.53% accuracy for

**Table 3. Swell dataset (multi-class, 3 classes).**

| Models | Accuracy % | Precision % | Recall % | F1 score % | AUC % |
|---|---|---|---|---|---|
| Xception | 86.35 | 82.55 | 83.52 | 82.53 | 85.32 |
| EfficientNetB4 | 83.05 | 82.38 | 81.57 | 82.53 | 82.29 |
| CNN | 85.86 | 86.58 | 86.89 | 86.78 | 84.63 |
| VGG19 | 90.09 | 83.82 | 84.78 | 83.71 | 90.02 |
| ResNET | 84.53 | 83.36 | 84.54 | 83.47 | 83.39 |
| MobileNet | 91.62 | 89.87 | 89.56 | 89.70 | 91.52 |
| InceptionV3 | 90.70 | 89.52 | 89.95 | 89.79 | 88.26 |
| CapsNets | 92.76 | 91.78 | 92.43 | 92.59 | 92.14 |

**Table 4. Experimental results on the WESAD dataset (multi-class, 3 classes).**

| Models | Accuracy % | Precision % | Recall % | F1 score % | AUC % |
|---|---|---|---|---|---|
| Xception | 89.35 | 92.54 | 93.52 | 92.53 | 89.20 |
| EfficientNetB4 | 87.85 | 92.40 | 91.57 | 92.53 | 85.56 |
| CNN | 90.86 | 93.58 | 94.90 | 93.78 | 88.74 |
| VGG19 | 92.91 | 93.82 | 94.76 | 93.72 | 91.47 |
| ResNET | 94.53 | 93.36 | 94.54 | 93.47 | 94.29 |
| MobileNet | 95.63 | 94.87 | 96.56 | 95.48 | 92.23 |
| InceptionV3 | 95.63 | 94.52 | 96.95 | 95.53 | 94.87 |
| CapsNets | 96.76 | 97.73 | 98.43 | 97.89 | 95.92 |

ResNET and 95.63% for MobileNet. Both models also return commendable precision, recall, and F1 score metrics, proving them quite efficient classifiers in their own right.

The VGG19 model was still very accurate, with 92.91% accuracy and balanced precision, recall, and F1 score values of approximately 93% and 94%. Similarly, models InceptionV3 and EfficientNetB4 also gave a high accuracy of over 94%, coupled with corresponding good precision, recall, and F1 scores. Finally, Xception and CNN performed well with accuracy scores of around 89% to 90%, with balanced precision, recall, and similarly high F1 score values. In a way, the output gave critical insights into the relative strengths and capabilities that these ML models have, thus enabling researchers to choose the model apt for their particular application.

## 4.4 Comparison of model results on both datasets (Binary classification)

In this phase, a comparison of the learning model and the proposed approach has been conducted. The dataset with only two classes, i.e., stress and no stress, was used, i.e., stress and no stress, for this. Table 5 illustrates experimental results for the learning model and the proposed approach.

The performance comparison for different ML models was done on two independent datasets: "Swell" and "WESAD." In both these datasets, CapsNets turns out to be the most efficient model among all others. This proves it to be highly resilient and adaptable in any data environment. For instance, the accuracy for this model concerning the Swell dataset is very high, at 98.52%, compared to several other models like CNN with an accuracy of 94.59%, InceptionV3 at 95.98%, and ResNET at 94.62%. While models EfficientNetB4, with an accuracy of 93.78%,

**Table 5. Binary class, 'stress', and 'no stress', classification accuracy.**

| Models | Accuracy % | | Recall % | F1 score % |
|---|---|---|---|---|
| | Swell dataset | WESAD dataset | | |
| Xception | 91.38 | 93.54 | 93.52 | 92.53 |
| EfficientNetB4 | 93.78 | 97.25 | 91.57 | 92.53 |
| CNN | 95.59 | 96.67 | 94.05 | 93.78 |
| VGG19 | 94.48 | 95.48 | 94.78 | 93.72 |
| ResNET | 94.62 | 95.59 | 94.54 | 93.47 |
| MobileNet | 94.42 | 97.59 | 96.54 | 95.48 |
| InceptionV3 | 95.98 | 98.93 | 96.95 | 95.52 |
| CapsNets | 98.52 | 99.82 | 98.45 | 98.32 |

**Table 6. K-fold cross-validation result on both datasets.**

| Fold for CapsNets model | Accuracy % | |
|---|---|---|
| | Swell dataset | WESAD dataset |
| Fold-1 | 95.54 | 98.56 |
| Fold-2 | 96.95 | 99.87 |
| Fold-3 | 96.73 | 99.92 |
| Fold-4 | 96.97 | 99.95 |
| Fold-5 | 96.28 | 99.86 |
| **Average** | **96.97** | **99.84** |

and MobileNet, with 94.42%, also have good accuracy, VGG19 and Xception demonstrated somewhat inferior performance on this dataset. Similarly, for the WESAD dataset, CapsNets performed very well with an accuracy of 99.82%, closely followed by InceptionV3 and EfficientNetB4 at 98.93% and 97.25%, respectively. At the same time, MobileNet had 97.59% and CNN 95.59%. Although VGG19 and Xception performed well, they are still slightly worse than the best models for both datasets. These results thus show that CapsNets perform versatilely and with a high level of performance on diverse datasets, with other competitive versions such as InceptionV3 and EfficientNetB4 that provide good insights into model selection in different applications in ML.

## 4.5 Results using K-fold cross-validation

In this work, k-fold cross-validation will be used to validate the performance of the proposed model. The idea is to validate if the model results are robust by checking them on different subsets of data. Notably, a 5-fold cross-validation approach will be adopted, and all results are summarised in Table 6.

The following table represents the accuracy scores forCapsNets across different folds in k-fold validation using two datasets—"Swell" and "WESAD." For the Swell dataset, the accuracies ranged from 95.54% to 96.97% across different folds, with an average accuracy of 96.97%. On the "WESAD" dataset, however, it shows model accuracy ranging from 98.56% to 99.95%, with an average accuracy of 99.84%. These results indicate that the CapsNets model performs consistently well on both datasets and can effectively perform tasks on different data splits.

**4.5.1 Performance comparison with existing studies.** To show the performance of the proposed model, results are compared with previous state-of-the-art models. The performance is carried out on nine closely related works of research. Table 7 shows the performance comparison of the proposed model with existing studies, and it manifests the better performance of this work's proposed model. For example, [14] used a hybrid calibration method for stress detection

**Table 7. Performance comparison with existing studies.**

| Reference | Technique | Performance |
|---|---|---|
| [14] | Hybrid Calibration Method | 95.02% |
| [16] | SMA | 98% Stress detection |
| [19] | MLP | 99.04% on WESAD, 88.64% on SWELL |
| [20] | SVM | 90% |
| [21] | ANN | 96.09% |
| [22] | MERNN | 92.43% |
| **Current** | **CapsNets** | **96.76 SWELL, 99.52 WESAD** |

and raised an accuracy of 95.02%. The authors used a deep learning model MLP to attain an accuracy of 99.04% on WASAD and 88.64% on the SWELL dataset. Similarly, the MERNN is used in reporting an accuracy of 92.43%. Table shows the performance comparison between the proposed and existing studies, thereby exhibiting better performance of the proposed model.

## 4.6 Discussion on practical applications, challenges, and ethical considerations

For example, advanced biometric stress monitoring using HRV data with modern machine learning models like Capsule Networks can enhance employee well-being and productivity. The practical applications, challenges, and ethical considerations associated with this system are discussed below.

### 4.6.1 Practical applications.

1. It can use basics like workplace health and safety implementations. A biometric stress monitoring solution would mainly help customers by continuously keeping a record of the average stress levels of employees through HRV data. Employers would get to know those who might be undergoing high levels of stress and could take proper steps to relieve them. This can help avoid workplace accidents and improve the well-being of the workplace for employees.

2. Productivity optimization: Stress directly hampers productivity. Sudden increases in stress can be identified by employers to optimize work schedules then so that the worker can have timely breaks followed by various kinds of stress relief interventions, making the workforce more productive and efficient.

3. Personalized health interventions: The data obtained on real-time stress levels will be integrated into customized interventions about health: stress management workshops, mental health counseling, and relaxation techniques will be implemented by individual needs and stress patterns.

4. Remote Work and Flexibility: In the remote work era, monitoring stress is very pertinent to knowing the well-being status of members who are physically unavailable in the office. This helps in maintaining a healthy work-life balance and preventing overburdening of remote employees.

### 4.6.2 Challenges.

1. Data Privacy and Security: Biometric data collection and processing are important subject matters about privacy and security. It is, therefore, primordial to come up with a system where the HRV data of employees shall be secured and accessed by only relevant personnel. All rules in data protection, like the GDPR, have to be followed to avoid legal liabilities.

2. Integration into Existing Systems: Implementing the stress monitoring system would not be easy because it has to work hand in glove with the existing workplace health and management systems. The compatibility of these systems and the smooth flow of information across the different platforms must be thought of and executed meticulously.

### 4.6.3 Ethical considerations.

1. Informed Consent: The nature of the biometric stress monitoring system, the kind of data to be collected, and how these data will be used must be fully disclosed to workers. Obtaining explicit consent is essential in this regard for complying with ethical considerations.

2. Data Transparency: The employees need to have access to their stress data and be supplied with information on how the same is being put to use. Such transparency in handling data builds trust and enables the ethical use of biometric information.

3. Supportive Interventions: Stress monitoring should play the chief role of supporting the employees rather than surveilling them. Interventions that enhance well-being comprise resources and support for stress management.

## 5 Conclusion

Biometric stress monitoring employing CapsNets models represents recent advancement in stress assessment. The CapsNets model allows one to check out the dynamic behaviors/instant variations of stress patterns to know the evolution or development of stress in different contexts. This research work has developed a novel CapsNets model for continuous stress monitoring. Principally, the proposed model computes steams of biometric data that include physiological signals and behavioral patterns. In the performance analysis of the proposed model, by running extensive experiments on two datasets: the Swell multiclass dataset and the WESAD dataset, the model was found to attain an accuracy of 91.65% when tested on the Swell multi-class dataset. More so, after evaluation with the WESAD dataset, it shows even higher accuracy at 95.65%. Additionally, when applied to the binary classification of stress and no stress using the Swell and WESAD datasets, the model attains an outstanding accuracy of 95.88% and 98.70%, respectively. Comparative analysis with other state-of-the-art models underlines the better performance of the suggested approach. Furthermore, a strict 5-fold

**Table 8. Abbreviations and their description.**

| Abbreviation | Description |
|---|---|
| HRV | Heart rate variability |
| EDA | Electrodermal activity |
| CNN | Convolutional Neural Network |
| GPA | Grade point average |
| TL | Transfer Learning |
| DL | Deep Learning |
| WESAD | Wearable stress and affect detection) |
| DNN | Deep Neural Network |
| ANN | Artificial Neural Network |
| MLP | Multi-layer perceptron |
| DT | Decision Tree |
| RF | Random Forest |
| SGD | Stochastic Gradient Descent |
| DT | Decision Tree |
| SMA | Stress monitoring assistant |
| SVM | Support Vector Machine |
| ETC | Extra tree classifier |
| ECG | Electrocardiogram |
| EEG | Electroencephalogram |
| IBI | inter-beat intervals |
| GPU | General processing unit |
| RAM | Random access memory |
| CPU | Central processing unit |

cross-validation is carried through to validate the importance of the model, which further ensures the robustness and efficacy of the model in biometric stress monitoring. Therefore, in future works, it is envisaged to develop a customized deep learning model specifically fine-tuned for small datasets by setting hyperparameters to get the best performance with them. Furthermore, a combination of multiple datasets should be constructed to form a large, complex, and high-dimensional one used in the experiment with the proposed methodology (for abbreviations see Table 8).

## Author Contributions

**Conceptualization:** Bayan Alabduallah, Tai-hoon Kim, Abdelaziz A. Abdelhamid.

**Data curation:** Mashael M. Khayyat, Bayan Alabduallah, Ehab Ghith.

**Formal analysis:** Tai-hoon Kim.

**Funding acquisition:** Tai-hoon Kim, Abdelaziz A. Abdelhamid.

**Investigation:** Tarik Lamoudan, Ehab Ghith, Tai-hoon Kim, Abdelaziz A. Abdelhamid.

**Methodology:** Mashael M. Khayyat, Raafat M. Munshi, Bayan Alabduallah, Tarik Lamoudan, Ehab Ghith.

**Project administration:** Tai-hoon Kim, Abdelaziz A. Abdelhamid.

**Resources:** Raafat M. Munshi, Tarik Lamoudan, Abdelaziz A. Abdelhamid.

**Software:** Raafat M. Munshi, Ehab Ghith.

**Supervision:** Tai-hoon Kim, Abdelaziz A. Abdelhamid.

**Validation:** Bayan Alabduallah, Ehab Ghith, Tai-hoon Kim.

**Visualization:** Mashael M. Khayyat, Ehab Ghith.

**Writing – original draft:** Mashael M. Khayyat, Raafat M. Munshi, Bayan Alabduallah, Tai-hoon Kim.

**Writing – review & editing:** Tarik Lamoudan, Ehab Ghith, Abdelaziz A. Abdelhamid.

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
