## [Decision Letter · Decision Letter 0]

19 Jun 2024

PONE-D-24-18697An Improved Biometric Stress Monitoring Solution For Working Employees Using Heart Rate Variability Data and Capsule Network ModelPLOS ONE

Dear Dr. Munshi,

Thank you for submitting your manuscript to PLOS ONE. After careful consideration, we feel that it has merit but does not fully meet PLOS ONE’s publication criteria as it currently stands. Therefore, we invite you to submit a revised version of the manuscript that addresses the points raised during the review process.

We look forward to receiving your revised manuscript.

Kind regards,

Mohamed Hammad, Ph.D.

Academic Editor

PLOS ONE

Journal Requirements:

Reviewers' comments:

Reviewer's Responses to Questions

**Comments to the Author**

1. Is the manuscript technically sound, and do the data support the conclusions?

Reviewer #1: Yes

Reviewer #2: Yes

2. Has the statistical analysis been performed appropriately and rigorously? 

Reviewer #1: Yes

Reviewer #2: Yes

3. Have the authors made all data underlying the findings in their manuscript fully available?

Reviewer #1: Yes

Reviewer #2: Yes

4. Is the manuscript presented in an intelligible fashion and written in standard English?

Reviewer #1: Yes

Reviewer #2: Yes

5. Review Comments to the Author

**Reviewer #1:** Strengths:

- Comprehensive literature review and context setting.

- Clear description of the methodology and experimental setup.

- Impressive accuracy rates and robust validation through k-fold cross-validation.

- Comparative analysis with state-of-the-art models.

Areas for Improvement:

- The introduction could be more concise with a clearer focus on the study's objectives.

- The related work section, while thorough, could benefit from a summary table or diagram.

- More detailed explanations of preprocessing and feature extraction methods in the methodology section.

- Inclusion of additional performance metrics (precision, recall, F1-score) in the results.

- Expanded discussion on practical applications, challenges, and ethical considerations.

**Reviewer #2:** This paper is well-written; however, a few improvements are needed:

English Corrections: Please review the text for any grammatical or typographical errors.

Additional Details: Please provide more information about the results. For instance, specify which kernel was used for the SVM and include the TensorFlow version, among other relevant details.

6. PLOS authors have the option to publish the peer review history of their article (what does this mean?). If published, this will include your full peer review and any attached files.

Reviewer #1: No

Reviewer #2: **Yes: **Krishna Mohan Kudiri

---

## [Author Response · Author response to Decision Letter 0]

28 Jun 2024

We have provided a separate PDF to address the reviewers' minor concerns.

---

## [Decision Letter · Decision Letter 1]

19 Jul 2024

PONE-D-24-18697R1An Improved Biometric Stress Monitoring Solution For Working Employees Using Heart Rate Variability Data and Capsule Network ModelPLOS ONE

Dear Dr. Munshi,

Thank you for submitting your manuscript to PLOS ONE. After careful consideration, we feel that it has merit but does not fully meet PLOS ONE’s publication criteria as it currently stands. Therefore, we invite you to submit a revised version of the manuscript that addresses the points raised during the review process.

We look forward to receiving your revised manuscript.

Kind regards,

Subramani Neelakandan

Academic Editor

PLOS ONE

Journal Requirements:

Additional Editor Comments:

Recent existing survey works must be improved by including the overall analysis as separate table. Author need to include some effective validation techniques of the proposed system

Please address the reviewer comments properly and do proofread for whole manuscript.

Reviewers' comments:

Reviewer's Responses to Questions

**Comments to the Author**

1. If the authors have adequately addressed your comments raised in a previous round of review and you feel that this manuscript is now acceptable for publication, you may indicate that here to bypass the “Comments to the Author” section, enter your conflict of interest statement in the “Confidential to Editor” section, and submit your "Accept" recommendation.

Reviewer #2: All comments have been addressed

2. Is the manuscript technically sound, and do the data support the conclusions?

Reviewer #2: Yes

3. Has the statistical analysis been performed appropriately and rigorously? 

Reviewer #2: Yes

4. Have the authors made all data underlying the findings in their manuscript fully available?

Reviewer #2: Yes

5. Is the manuscript presented in an intelligible fashion and written in standard English?

Reviewer #2: Yes

6. Review Comments to the Author

Reviewer #2: This paper is very well written. All the comments were addressed by the author however, please re-check the data representation in the tables. For instance, Table no 6 has Accuracy, it should be like " Accuracy (%)".

7. PLOS authors have the option to publish the peer review history of their article (what does this mean?). If published, this will include your full peer review and any attached files.

Reviewer #2: **Yes: **Krishna Mohan Kudiri

---

## [Author Response · Author response to Decision Letter 1]

20 Jul 2024

We have provided a separate PDF to address the reviewer 2 minor concerns.

---

## [Editor Report · Decision Letter 2]

16 Aug 2024

PONE-D-24-18697R2An Improved Biometric Stress Monitoring Solution For Working Employees Using Heart Rate Variability Data and Capsule Network ModelPLOS ONE

Dear Dr. Munshi,

Thank you for submitting your manuscript to PLOS ONE. After careful consideration, we feel that it has merit but does not fully meet PLOS ONE’s publication criteria as it currently stands. Therefore, we invite you to submit a revised version of the manuscript that addresses the points raised during the review process.

**ACADEMIC EDITOR: **Author need to do complete proof read for entire manuscript. for example "The accuracy obtained for binary classification of stress vs. no stress is applied to the Swell dataset, where this model obtained an outstanding accuracy of 98.52% in this study and on WESAD, 99.82%. Here how to understand vs ? I mean it is a versus however the meaning is entirely different similarly I can found many statement describing different meaning. therefore author need to do complete proof read for entire manuscript. In table.1 author discussed review summary table, Fig.1 showing the flow from dataset to SWELL then WESAD. here WELL and WESAD both are dataset then why we need to point these direction ? If the input is SWELL and WESAD is the dataset then why these three steps. Also what is the pre-processing steps followed any why 85:15 ration for trains and testing process ? How it has been validated ? Many abbreviations not defined properly when  it use first time int he manuscript for example SGD, ECG, SVM etc... Dataset description can be discussed in experimental setup information. Also its better author can include the fitness evaluation of the model. whether any hypothesis testing was conducted ? if not please include the testing results. Capsnets gained 98.52 and 99.82% accuracy than InceptionV3 is really good if the author can share the implementation works details in GitHub and cite the reference link in the abstract.  

We look forward to receiving your revised manuscript.

Kind regards,

Subramani Neelakandan

Academic Editor

PLOS ONE
---

## [Editor Report · Decision Letter 3]

6 Sep 2024

An Improved Biometric Stress Monitoring Solution For Working Employees Using Heart Rate Variability Data and Capsule Network Model

PONE-D-24-18697R3

Dear Dr. Munshi,

We’re pleased to inform you that your manuscript has been judged scientifically suitable for publication and will be formally accepted for publication once it meets all outstanding technical requirements.

Kind regards,

Subramani Neelakandan

Academic Editor

PLOS ONE

Additional Editor Comments (optional):

All the Comments addressed by the authors.

---

## [Editor Report · Acceptance letter]

27 Sep 2024

PONE-D-24-18697R3 

PLOS ONE

Dear Dr. Munshi, 

I'm pleased to inform you that your manuscript has been deemed suitable for publication in PLOS ONE. Congratulations! Your manuscript is now being handed over to our production team.

Kind regards, 

on behalf of

Dr. Subramani Neelakandan 

Academic Editor

PLOS ONE